# Influence of grain size on the solid-state direct reduction of polycrystalline iron oxide
Barak Ratzker [1] ✉, Martina Ruffino[1], Shiv Shankar [1], Yan Ma [1,2] & Dierk Raabe[1]

Direct reduction of iron oxide using hydrogen offers a sustainable route to lower carbon emissions in steelmaking. Although iron oxide feedstocks consist of polycrystalline pellets, the influence of initial hematite grain size on direct reduction remains unexplored. Herein, the effect of grain size on reduction kinetics and microstructure evolution were uncovered using model polycrystalline hematite samples with large (~ 30 μm) and ultrafine (~ 1 μm) grains. Thermogravimetric analysis showed grain-size-dependent reduction behavior, while microstructural examination of partially reduced samples revealed that large-grained hematite forms finer directional pore channels due to fewer grain boundaries and orientation changes. Consequently, large-grained samples reduce faster initially as the pore network develops, while ultrafine-grained samples achieve more efficient reduction in later stages facilitated by a more homogenous pore network. These results demonstrate how grain size dictates porosity and texture evolution, providing fundamental insights relevant not only to hydrogen-based iron production but also to the design of porous materials by solid-state reduction processes.

Primary iron production from oxides relies mostly on unsustainable carbon-based reduction pathways using fossil reductants, making it one of the most energy intensive sectors and one of the largest single sources of global anthropogenic $CO_2$ emissions (roughly 8% of the total)[1]. Thus, there is increasing effort to produce iron for green steel via alternative, more sustainable routes such as hydrogen-based direct reduction[2,3]. These processes rely on direct reduction of iron oxide prepared from ores[4] using hydrogen gas as the reductant, which releases $H_2O$ as the byproduct and thereby offers a route towards carbon neutrality[5]. Direct reduction is a microstructurally intricate solid-state process that involves multiple sequential phase transformations alongside concurrent solid and gas transport mechanisms[6]. Therefore, widespread implementation of sustainable direct reduction for iron production could be facilitated by gaining better understanding of microstructure evolution during this process.

Many factors influencing the kinetics and microstructure evolution of iron oxide direct reduction have been investigated[7]. Examples include the effects of reducing conditions such as temperature[8,9], gas pressure[10,11], gas flow rate[10], reductant gas composition[12], and material-related properties like initial porosity[13], commercial pellet type/composition[14], particle size[15,16], gangue oxides[17], and chemical impurities[18], among others. However, although some past studies may have erroneously equated 'grain size' with powder particle size[7,19], the specific effect of grain size on the direct reduction of polycrystalline iron oxide has not yet been examined. It should be clarified that herein the term 'grains' only refers to the crystallites comprising the polycrystalline oxide outlined by grain boundaries – and not what is sometimes in the literature referred to as "grains" in direct reduction kinetics grain models[20,21], which are the localized core-shell units that develop during reduction, consisting of an oxide core enveloped by lower oxide states or metal and surrounded by large pores.

Despite the extensive research on direct reduction of iron oxide, including many microstructure-oriented studies[9,11,18,22–26], there are still microstructural aspects that have been overlooked – most notably the effect of hematite grain size. It is well-established that grain size (as determined by the density of grain boundaries separating regions of the material where the crystal lattice is continuous and oriented in the same direction) is a dominant factor in relation to polycrystalline materials that influence a wide range of mechanical[27–29], functional[30–32], and transport[33–36] properties. Since the direct reduction process involves multifaceted transient microstructural mechanisms such as diffusion, phase transformations, strain, vacancy formation, and void coalescence – it is expected that grain size would influence the iron oxide direct reduction kinetics and efficiency by affecting the microstructure evolution.

The microstructure development during solid-state direct reduction, especially in dense materials, proceeds in a pronounced topochemical manner[37,38], with an advancing reduction front parallel to the outer surfaces. This macroscale behavior is commonly described as a shrinking core

[1]Max Planck Institute for Sustainable Materials, Düsseldorf, Germany. [2]Department of Materials Science and Engineering, Delft University of Technology, Delft, The Netherlands. ✉e-mail: b.ratzker@mpi-susmat.de

**Table. 1 | Physical properties and reduction degrees of polycrystalline hematite samples with large (AS) and ultrafine (SPS) grain sizes**

| Sample sintering process | Initial relative density (%) | Mean grain size (µm) | Reduction degree (%) | | | |
|---|---|---|---|---|---|---|
| | | | 2 °C/ min (30 min)[a] | 10 °C/ min (30 min)[a] | 20 °C/ min (30 min)[a] | 10 °C/ min (1 min)[a] |
| SPS at 900 °C for 1 h | ~99.5 | 1.0 ± 0.4 | 99.3 | 94.0 | 93.2 | 27.6 |
| AS at 1200 °C for 4 h | ~98 | 34.5 ± 14.7 | 99.2 | 91.8 | 86.5 | 42.3 |

[a]Isothermal holding time at 700 °C under 100% $H_2$ flow.

model[20,39]. Once there is interconnected porosity for gas transport, reduction ensues more uniformly throughout the entire material by localized micro-scale core-shell formations with oxides being encased in dense product metal, typically referred to as a grain or pore model behavior[20]. In our recent study on the hydrogen-based direct reduction behavior of single crystal hematite[26], it was shown that the reduction process can be described microstructurally by the combination of the 'shrinking-core' and 'pore/grain' models. It is unknown how the presence of grain boundaries influences either of these regimes or the overall kinetics.

An important aspect that relates to grain size is how it impacts porosity formation during the direct reduction process and the ensuing development of the nano- and microscale pore networks[40]. This induced porosity that continuously evolves plays a critical role in furthering reduction by creating free surfaces for reaction and facilitating the evacuation of water vapor[18,41,42]. Moreover, beyond the role of pores in iron production, porosity volume fractions, morphology, and connectivity could be crucial factors for optimizing performance and cyclic behavior in energy-related applications that utilize iron oxide and can benefit from controlled porosities and degradation minimization, such as metal-air batteries[43,44], metal oxide fuel combustion[45–47], and carbon capture materials[48,49], as well as tailoring the design of porous alloys/composites produced from sintered mixed oxide precursors[50,51].

The iron oxide feedstock used in the iron production industries consists of porous, polycrystalline, sintered spherical pellets (roughly 0.9–1.2 cm in diameter) comprising mostly (>90%) hematite ($Fe_2O_3$), alongside minor fractions of other iron oxides like magnetite ($Fe_3O_4$) and common gangue oxides like $SiO_2$, $Al_2O_3$, CaO, and MgO[7]. Given the microstructural complexity of industrial pellets and their inhomogeneous grain size[18,52], we chose to use sintered hematite simulants, as done in past studies for other purposes[37,39,53–57]. Moreover, altering the sintering conditions provides the opportunity to fabricate pure, dense polycrystalline hematite with varying grain sizes to directly study the isolated effect of grain size during direct reduction.

In the present study, we aim to uncover the effect of grain size on direct reduction behavior of iron oxide by focusing on a comparison between two sintered hematite materials with considerably different grain sizes: large (~30 µm) and ultrafine (~1 µm) grains, which approximately represent the low and high end of grain sizes typically found in commercial hematite pellets[18,52]. Complete and partial reduction was performed under 100% $H_2$ atmosphere at 700 °C, enabling us to assess the direct reduction kinetics and microstructure evolution. Through this approach, we isolate and determine how the hematite grain size affects the solid-state direct reduction process. It was found that initial hematite grain size predominantly governs the pore network formation, with large grains developing finer but more heterogenous porosity while in fine-grained samples the pores are comparatively coarse but more homogenous. In turn, the pore network morphology dictates the overall reduction behavior including kinetics, efficiency, and final product microstructure.

## Results and discussion
### Direct reduction kinetics
Sintered hematite samples were produced using submicron hematite powder (Supplementary Fig. 1). Spark plasma sintering (SPS) yielded samples with an *ultrafine* grain size of ~1 µm in contrast to air sintered (AS) samples with a *large* grain size of ~30 µm (see analysis of the grain sizes and sinter pores in

Supplementary Figs. 2, 3). The initial porosity in all sintered samples was very low (≤~2%), so that the isolated sinter pores have a negligible effect on the reduction behavior. Table 1 presents a comparison of physical properties and the reduction degrees achieved for the two kinds of samples. The hydrogen-based direct reduction kinetics of samples with different grain sizes were assessed by thermogravimetric analysis (TGA) at 700 °C (Fig. 1). Figure 1a, b shows the reduction degree ($R$) and reduction rate ($dR/dt$) as a function of time, respectively. Further, Fig. 1c shows the reduction rate as a function of reduction degree. By inspecting the latter, some notable differences can be observed: at the earlier reduction stages (~10–35% reduction) the large grain sample reduced more quickly than the ultrafine-grained sample; in contrast, for most of the remainder of the reduction process (~45–90% reduction) the ultrafine-grained sample reduced more quickly, especially as the reduction rate peaked between 60 and 65% reduction degree.

Figure 1d shows the appearance of the sintered hematite samples before and after the direct reduction process. The reduced samples present visible shrinkage with their diameter decreasing by roughly ~5%. This shrinkage contrasts with the swelling that has been observed in sintered hematite reduced with CO gas[58]. Furthermore, despite the shrinkage the samples are still quite porous, with the large- and ultrafine-grained samples exhibiting similar total porosity volume fractions of about ~38% and ~35%, respectively, 90% of which being interconnected open porosity, in agreement with typical porosity levels found in direct reduced iron ore[59]. All types of samples exhibited extensive surface cracking following reduction (see Supplementary Fig. 4a). The additional rupturing in the ultrafine-grained sample is attributed to the residual stress remaining after the pressure-assisted sintering. Although the addition of exposed surfaces would be thought to enhance reduction kinetics, it is more likely to have a negligible effect on the overall reduction behavior, as this rupturing occurs only at later stages when reduction is not topochemical on the macroscale, but dominated by the relatively uniform "grain/pore model" reduction behavior through the percolating microporosity and microcrack network.

The effect of heating rate on the reduction behavior was also investigated. By the end of the programmed reduction process with a heating rate of 10 °C/min, the total reduction degree of the large-grained sample (92%) was slightly lower than the ultrafine-grained sample (94%). When comparing the total reduction degree as a function of various heating rates (for the same isothermal holding at 700 °C for 30 min) a clear trend emerges: at high heating rate (20 °C/min) the difference in reduction degree between the large- and ultrafine-grained hematite noticeably increases, whereas at low heating rate (2 °C/min) both samples exhibit nearly the same very high reduction degree, >99%. Thus, the hematite with large grain size seems more susceptible to having a larger fraction of retained unreduced oxides at higher heating rates.

### Microstructure analysis
Microstructural analysis was carried out on partially reduced samples held at 700 °C for 1 min. After being subjected to the same reducing conditions there was a considerable difference in reduction degree between the samples: the large and ultrafine grain size samples reduced 42.3% and 27.6%, respectively. This behavior agrees with the reduction kinetics observations and the underlying reasons can be revealed by analyzing the reduced microstructures. The samples were cut to expose the circular cross section (Supplementary

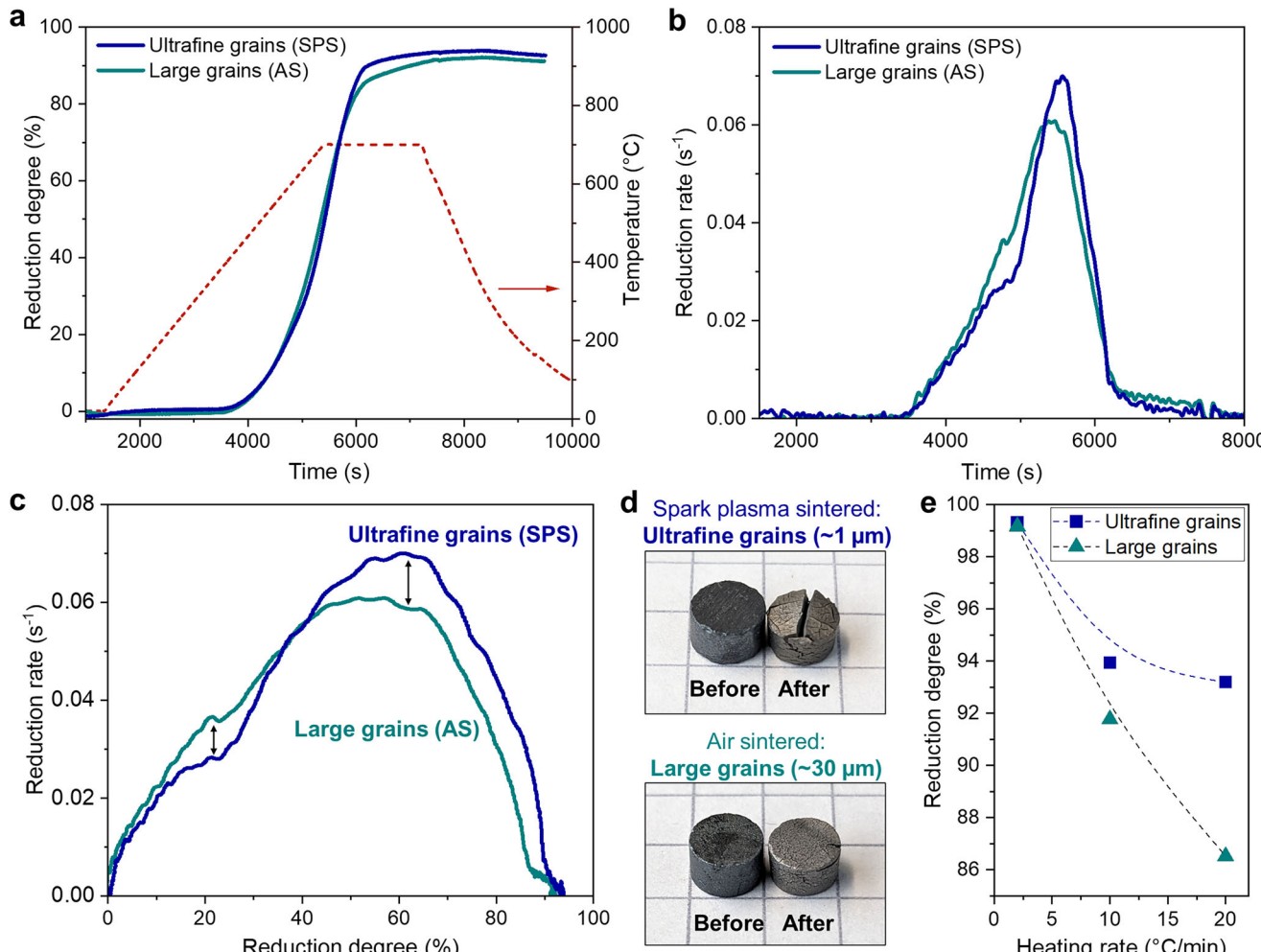

**Fig. 1 | Direct reduction kinetics.** Comparison of reduction kinetics of ultrafine-(~1 μm) and large-grained (~30 μm) sintered hematite samples investigated by thermogravimetric analysis (TGA). Experiments were carried out with 100% $H_2$ with a heating rate of 10 °C/min up to 700 °C, followed by isothermal holding for 30 min. **a** Reduction degree as a function of time alongside the heating profile temperature. **b** Reduction rate as a function of time. **c** Reduction rate as a function of reduction degree. **d** Appearance of the samples before and after the direct reduction process (placed on a regular math page where each square edge is 5 mm for reference scale). **e** Total reduction degrees of ultrafine- and large-grained samples as a function of heating rate (2, 10, and 20 °C/min).

Fig. 4b), revealing a reduced outer layer and unreacted core. As shown in Fig. 2, the reduction proceeds in a distinct shrinking-core topochemical manner – as expected for dense hematite[26,37]. Figure 2a, b shows the noticeable macroscale differences between the sample with the large grain size (~30 μm) and that with the ultrafine grain size (~1 μm). The large-grained sample exhibits a ~30% thicker reduced layer (~390 μm compared to ~300 μm for the ultrafine-grained sample) and its reduced layer also appears to contain more metallic iron (based on the brighter BSE contrast). This can account for the substantial difference in reduction degree between the two samples. Upon closer inspection near the interface with the unreacted hematite (Fig. 2c, d), a layer of magnetite can be seen separating the hematite and the reduced layer containing a mixture of wüstite and metallic iron in both samples. However, the ultrafine-grained sample exhibits a more homogenous microstructure within the reduced layer, regarding the distribution of oxide and metal phases as well as porosity.

The changes in oxide/metal phases and porosity fractions across the developing reduction front were assessed by image analysis. Supplementary Fig. 5 presents compound SEM-BSE images of a slice encompassing nearly the entire reduced layer in both samples, showcasing the microstructure evolution sequence as reduction progresses. Figure 3 presents the compositional profiles for both samples, starting at the hematite/magnetite interface and reaching up to ~250 μm into the reduced layers. Some similarities and differences can be identified between the two samples. There is a similar monotonous decrease in oxide phases and increase in the porosity and iron fractions up to ~60 μm from the interface with all the fraction values being quite similar. From that distance onwards, the porosity fraction in the SPS ultrafine-grained sample is higher than in AS large-grained sample by roughly 5–10%, remaining so for the remainder of the analyzed range. Despite the inherent limitations of using 2D analysis to describe a 3D pore structure, the porosity fractions are assumed to be adequately representative[60]. Additionally, the metallic iron fraction in the large-grained sample continues to increase while that in the ultrafine-grained sample stabilizes, such that iron content in the large-grained sample quickly becomes ~15–20% higher than that in the ultrafine-grained sample for the next ~150 μm. The higher iron content in the partially reduced large-grained sample is in agreement with the higher reduction degree recorded by mass loss. On the other hand, the oxide fractions retain rather similar trends in both samples throughout the reduced layer.

Figure 4 shows higher magnification of the hematite/magnetite interface and the nearly fully reduced regions close to the outer surface of the samples, revealing the porosity that develops during direct reduction. Note that the formation of fine interconnected pore networks in magnetite is a hallmark of direct reduction[9,22–24], facilitating the process in dense material. In the large-grained sample the pores are extremely fine (~10–100 nm in diameter), forming elongated and seemingly well-

**Fig. 2 | Overview of partially reduced microstructures.** BSE-SEM images of the reduced layer in the sintered hematite samples partially reduced for 1 min at 700 °C with a heating rate of 10 °C/min. **a** Large and **b** ultrafine grain size, the thickness of the reduced layers is indicated. The entire cross-section of the samples is shown in the respective insets. **c, d** Show zoom-in images showcasing the region closer to the hematite/magnetite interface marked by dashed rectangles in (**a**) and (**b**), respectively.

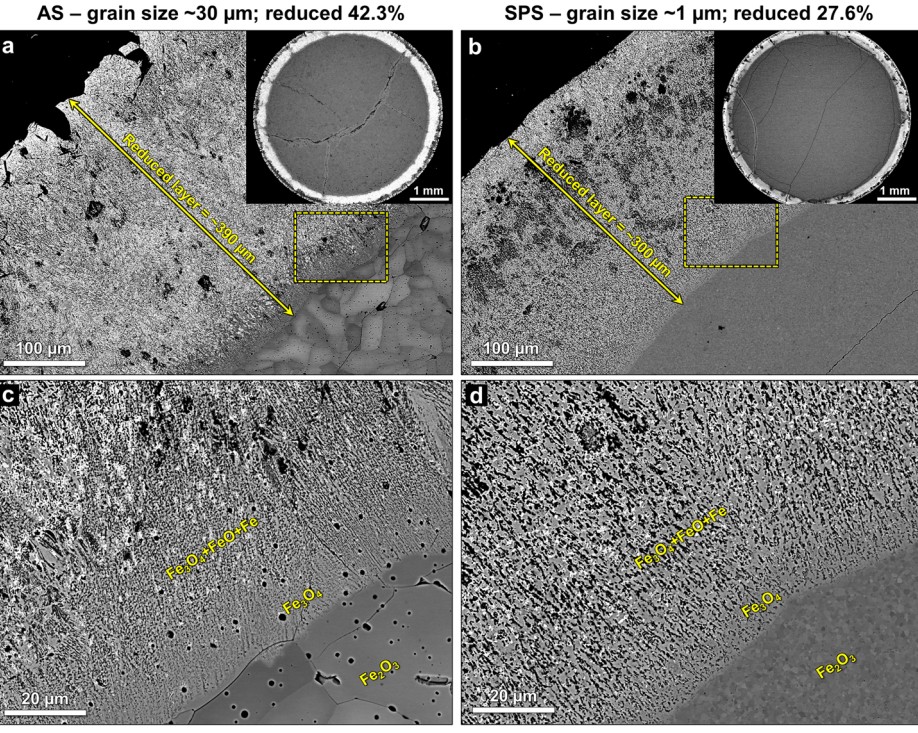

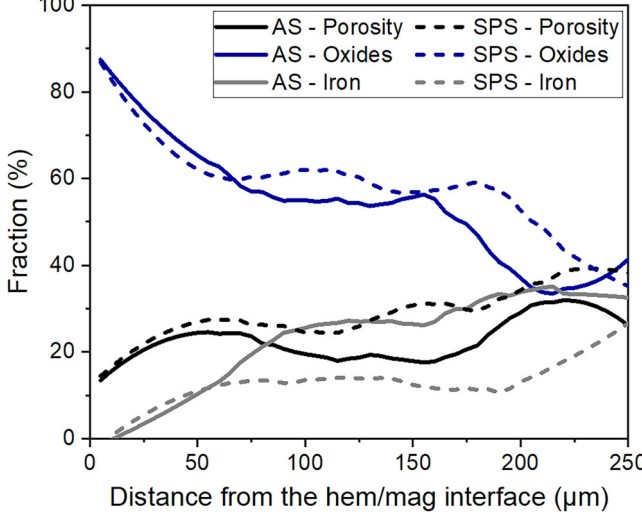

**Fig. 3 | Porosity and phase composition in the reduced layers.** Fractions of porosity, oxide, and iron measured across the reduced layers starting from the hematite/magnetite interface up to ~250 μm into the reduced layer. The solid and dashed lines represent the (AS) large- and (SPS) ultrafine-grained hematite samples, respectively.

connected nanoscale pore channels that extend from the hematite/magnetite interface deep into the reduced layer. This pore morphology closely resembles that which developed within the magnetite in single crystal hematite reduced under similar conditions[26]. Conversely, in the ultrafine-grained sample, the porosity at the reduction front is noticeably more spherical and relatively coarse (~100–500 nm in diameter). Pore connectivity also appears to be poorer in the magnetite near the hematite/magnetite interface.

Evidently, the small grain size of the hematite predecessor has a significant effect on the initial pore formation in magnetite, hindering the directional accumulation of vacancies that create very fine and typically straight channels in magnetite that formed from large-grained hematite. Higher magnification BSE-SEM images of the nanopores forming at the

hematite/magnetite interface are shown in Supplementary Fig. 6. Moreover, Supplementary Fig. 7 shows the same regions at lower magnification alongside image analysis of the pore network directionality with corresponding histograms depicting the angular distribution. Evidently, there is noticeable variability in the porosity directionality in the sample with large hematite grains, wherein different grains can display inferior or superior pore directionality. This suggests that crystallographic orientation likely determines the directionality of the initial pore channels within individual large grains. Nevertheless, even when there is poor pore network directionality, the pore channels formed within the magnetite in the large grain size samples are always considerably finer than those in the ultrafine-grained samples. The finer and more interconnected initial pore channel formation in the large-grained sample suggests faster progression of the advancing reduction front, explaining the higher reduction degree at earlier reduction stages. Furthermore, the fact that the fraction of porosity is larger than that of iron in the large-grained sample and vice versa for the ultrafine-grained sample (Fig. 3) can also be explained by their different pore network morphology: the more metallic iron forming during reduction, the more it sinters, compounded by the fact that since the channels are narrower in the large-grained sample, rapid sintering is promoted by stronger capillary forces and a higher proportion of surface to volume ratio (the driving force for sintering).

To further demonstrate how the pore network channel morphology (dimensions and directionality) depends on hematite grain size, the characteristic porosity in partially reduced large- and ultrafine-grained samples were compared to that of the two other intermediate grain sizes of ~5 μm and ~10 μm, as is exemplified in Supplementary Fig. 8. There is a noticeable trend in which the pore network channels are straighter and finer the larger the grain size. The porosity that developed in the ~5 μm sample is roughly spherical and not very directional, resembling that which is found in the ultrafine (~1 μm) sample, whereas the porosity found in the ~10 μm sample is evidently quite straight and directional, as seen in the large-grained (~30 μm) sample. Hence, there is apparently a threshold for hematite grain size above which the subsequent magnetite is able to form more directional narrower pore channels. This is an important observation because it proves that the pore network

**Fig. 4 | Reduced microstructures at initial and late stages of reduction.** BSE-SEM images of different representative regions in the hematite samples, partially reduced at 700 °C with a heating rate of 10 °C/min. The hematite/magnetite interface for **a** ultrafine and **b** large grain size samples. The highly reduced Fe-rich region taken ~30 μm away from the outer surface for the **c** ultrafine and **d** large grain size samples. The yellow arrowheads point to examples of retained oxides encased by iron.

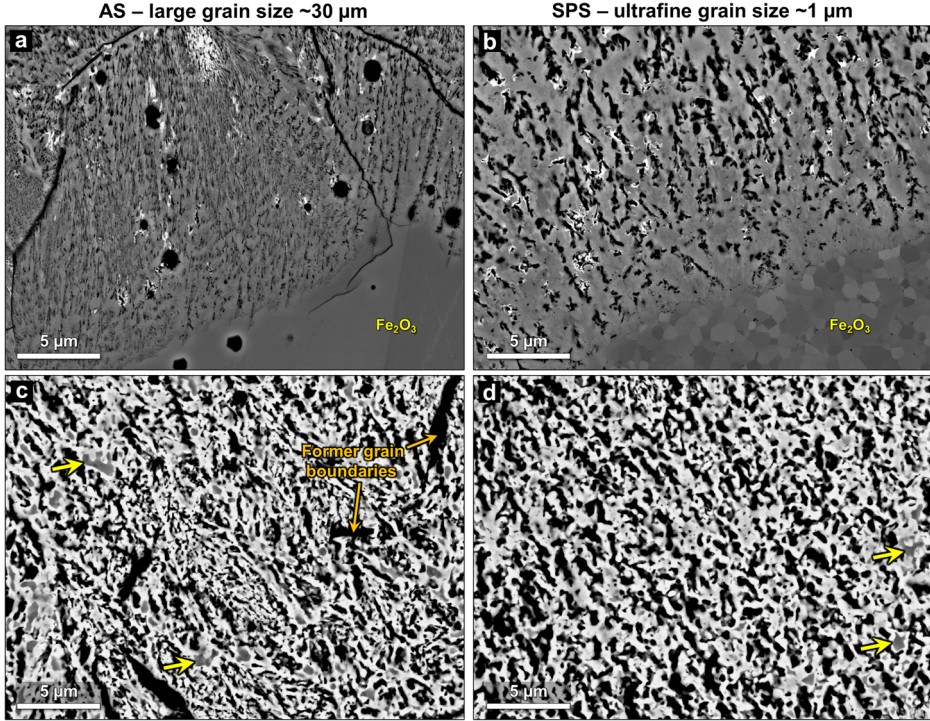

development depends on the grain size itself and not the grain boundary energy, which is typically lower in SPS samples that have more faceted grain boundaries close to optimal 120° dihedral angles.

Figure 4c, d shows the microstructures of Fe-rich regions near the outer surface of each sample. In these regions nearly all the oxide was reduced to iron, but differences in the porosity can still be observed. The pores in the ultrafine-grained sample are coarser and more uniform in size and shape compared to the porosity in the large-grained sample. In the latter, there is a high density of extremely fine pores (that are rarely observed in the other sample), as well as some larger elongated pores where hematite grain boundaries used to be. The directionality of the large pores does not necessarily follow the preferential outward radial direction shared by the fine pores, nor are they as well-connected to the rest of the pore network. Therefore, they may not be as effective in facilitating the evacuation of water vapor generated during the direct reduction process. Moreover, the extremely fine pore channels are probably more susceptible to sintering or reoxidation[42], which further slows down the final stages of reduction.

While there are residual oxides in both partially reduced samples, in the large-grained sample it is possible to occasionally find additional relatively oxide-rich regions (Supplementary Fig. 9). These retained oxides can be attributed to the more heterogeneous microstructure development during reduction; their slow reduction, driven by local core–shell behavior as the iron layer densifies, likely contributes to the decreased reduction rate at later stages and the lower final reduction degree at higher heating rates. (Fig. 1e). Therefore, it can be inferred that a large hematite grain size enhances the initial macroscopic "shrinking-core" reduction rate and enables faster progression of the reduction front into the bulk of the material but also creates a more heterogeneous microstructure. Consequently, the ensuing "grain/pore model" reduction – the more rate-limited step, since the conversion of wüstite to iron requires solid-state diffusion of oxygen through the surrounding metal – proceeds more quickly after finer hematite grains produce a more uniform microstructure with relatively coarser porosity and smaller retained oxide regions. Given the inconclusive similarities/differences in microstructure evolution observed for $H_2$ and CO reductants[9,23,61–63], it is unclear at this stage to what extent similar trends would occur when using other reducing gases like CO or $CH_4$.

EBSD analysis was conducted to gain deeper insights into the partially reduced microstructures in the samples with large and ultrafine hematite grain sizes. Figure 5 shows EBSD phase and inverse pole figure (IPF) maps of the cross sections of the reduction front at the hematite/magnetite interface and at the iron-rich, highly reduced regions. Examination of the reduction front near the hematite/magnetite interface (Fig. 5c, d) reveals the significant difference in phase, grain morphology, and texture evolution between the samples with ultrafine and large hematite grains. There is some grain refinement as the hematite transforms into magnetite even in the ultrafine-grained material. The magnetite grains near the interface are also visibly elongated along the direction of motion of the reduction front and exhibit very limited texture. In the large-grained sample, the magnetite grains close to the interface are of similar large proportions, elongated, and noticeably textured.

In both samples the first wüstite and iron grains formed at about ~5 μm away from the hematite. However, the thickness of the magnetite layer in the ultrafine-grained sample (~10 μm) is roughly half the thickness of that in the large-grained sample (~20 μm). Furthermore, in both samples there is almost no grain refinement during the magnetite to wüstite transformation due to the cube-on-cube relationship between the phases[64]. The retention of texture in the magnetite is what enables the formation of a finer straight pore channel network, which is possible if there are no grain boundaries, such as in single crystals[26] or if the grain size is large enough as shown here. In contrast, finer hematite grain size implies frequent orientation changes that prevent the formation of narrow directional porosity; more spherical, coarser pores are generated instead.

Some additional differences can be observed upon inspection of the iron-rich reduced regions. Many of the iron grains created from the large hematite grains are irregularly shaped and relatively large, containing sub-grains with low-angle grain boundaries (Fig. 5a). Furthermore, the residual oxides (mostly wüstite) are locally textured and elongated without high-angle grain boundaries (see Supplementary Fig. 10). In contrast, the iron grains created from the ultrafine hematite are equiaxed (Fig. 5b). Similarly, the residual oxides are also comprised of small, mostly equiaxed grains with no recognizable texture. The average iron grain size is smaller in the ultrafine-grained sample (0.34 ± 0.14 μm) compared with that of the large-grained sample (0.56 ± 0.41 μm). The corresponding grain size distributions are shown in Supplementary Fig. 11.

**Fig. 5 | Cross section EBSD analysis of micro-structures at initial and late stages of reduction.**
Reduced Fe-rich regions in the samples that initially had **a** large and **b** ultrafine hematite grains. An example of the relatively large irregular iron grains containing many subgrains found in the large-grained sample is circled in (**a**). Phase map and IPF on the left and right, respectively. The reduction front near the hematite/magnetite interface in **c** large- and **d** ultrafine-grained samples, phase map and IPF on the left and right, respectively. Note that the nanoscale pore networks cannot be resolved by EBSD.

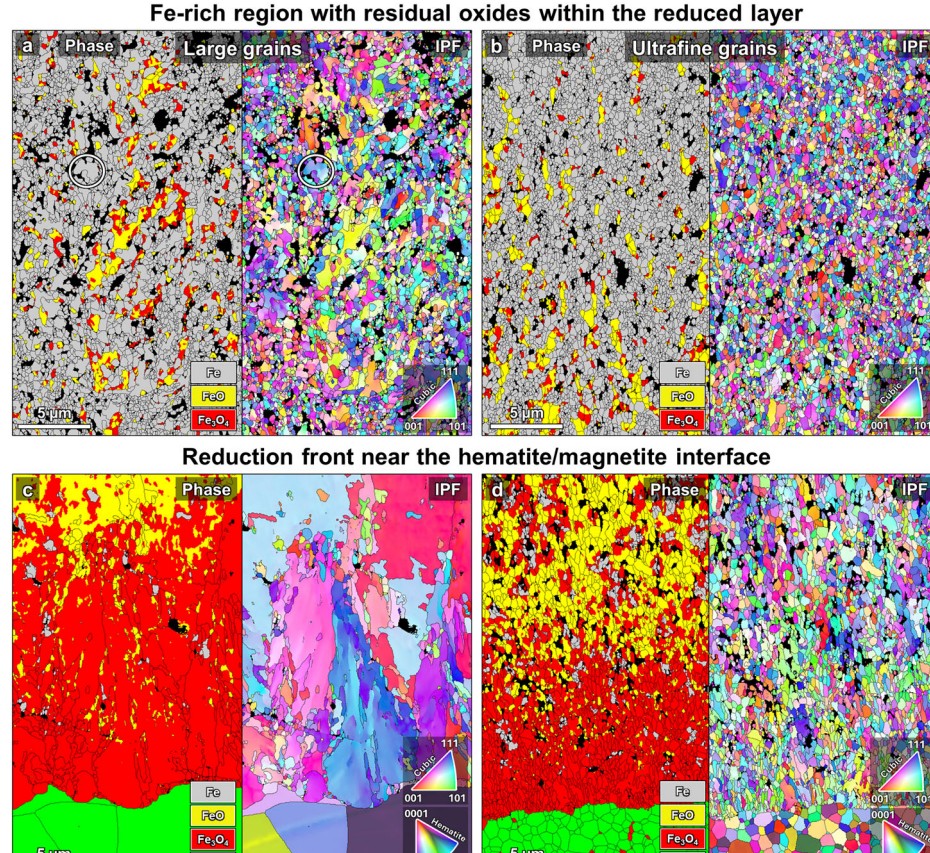

**Fig. 6 | EBSD analysis of the hematite/magnetite interface in partially reduced hematite with large grain size. a** Phase map with S-N type OR $(0001)_{Hem}\|(111)_{Mag}$ & $[1\bar{1}00]_{Hem}\|[1\bar{1}0]_{Mag}$ phase boundaries marked in blue and {111} 60° FCC twins in magnetite marked in cyan. **b** IPF map. The dark blue arrows indicate regions lacking the S-N OR leading to refinement of the magnetite, and the white arrows indicate protrusions of magnetite into the hematite where the S-N OR is obeyed.

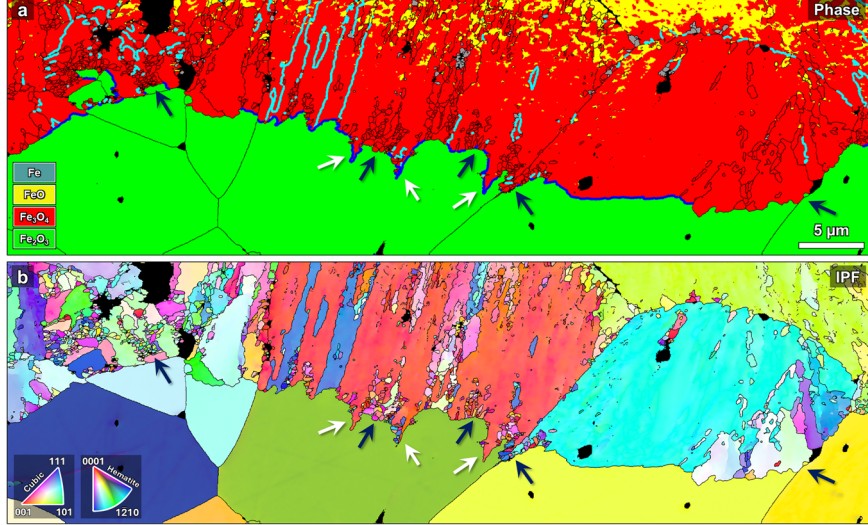

Figure 6 shows more in-depth analysis in the large-grained samples at the hematite/magnetite interface, where some interesting observations can be made. Figure 6a illustrates the prevalent $(0001)_{Hem}\|(111)_{Mag}$ & $[1\bar{1}00]_{Hem}\|[1\bar{1}0]_{Mag}$ Shoji-Nishiyama (S-N) orientation relationship (OR) observed between the hematite and magnetite phases[65]; the blue line in the phase map indicates boundaries separating regions that follow this OR. An interfacial region over 500 μm in length was analyzed by EBSD and a sizable fraction (~40%) of the interface separates hematite and magnetite regions with S-N OR. This is in contrast with observations of ORs in single crystal hematite reduced under similar conditions[26], where another known

orientation relationship[66] of $(0001)_{Hem}\|(112)_{Mag}$ & $[1\bar{1}00]_{Hem}\|[1\bar{1}0]_{Mag}$ (termed OR2) was found to be predominant; such regions are occasionally also present here, but are considerably less prevalent than S-N OR regions. The preferential development of such an OR is thought to depend on the relationship between the initial hematite orientation and the direction of the gradient of the oxygen chemical potential. For instance, OR2 was observed during reduction of single crystal hematite predominantly when the hematite *c*-axis was aligned with the direction of advancement of the reduction front[26]. Considering the multiplicity of orientations in poly-crystalline samples, the more common S-N OR is more likely to occur. On

**Fig. 7 | Microstructural analysis of the hematite/ magnetite interface as revealed by grinding into the reduced layer from the surface.** BSE-SEM images, EBSD phase and IPF maps of samples with (**a**, **c**, **e**) large and (**b**, **d**, **f**) ultrafine grain size, respectively. Notice the distinct "cell structure" morphology in the magnetite near the interface for the large-grained hematite, which is not discernible in the ultrafine-grained sample.

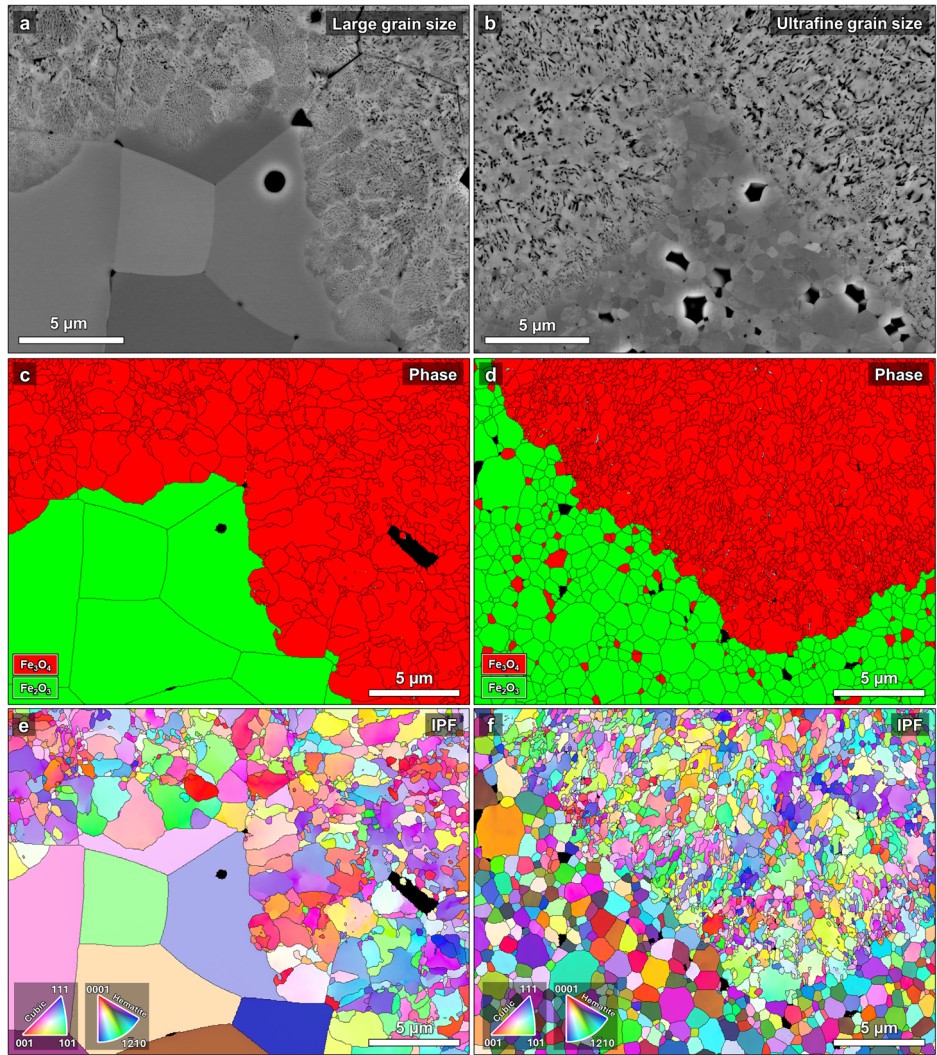

the other hand, in the ultrafine-grained sample there was no dominant orientation relationship observed. In some grains with the S-N OR, porous magnetite is seen to protrude into the hematite, marked by white arrows in Fig. 6. The cyan lines indicate {111} twin boundaries separating magnetite grains that nucleated as different crystallographic variants of S-N OR in the same hematite grain[67].

Another noteworthy observation for the hematite/magnetite interface in the large-grained hematite sample is the dependence of magnetite grain morphology on the orientation relationship with the hematite. Magnetite grains growing from hematite with S-N OR appear large and elongated, often spanning the entirety of the original hematite grain. In contrast, as is clearly indicated by the dark blue arrows in Fig. 6, where hematite and magnetite are not related by S-N OR there is substantial refinement of magnetite grains (typically by more than an order of magnitude) into ultrafine non-textured grains. These extreme morphological differences in magnetite formation point to different phase transformation pathways, likely involving atomistic mechanisms influenced by the interface orientation relationship.

To fully understand the microstructure development at the hematite/magnetite interface the samples were analyzed after grinding from the surface into reduced layer – i.e. in the direction which is perpendicular to the cross section and parallel to the direction of motion of the reduction front (see schematic depiction in Supplementary Fig. 4). This analysis is complementary to that conducted in cross-sectional view above, allowing for 3D understanding of the microstructure. Figure 7 shows SEM and EBSD maps of the ground-in sample hematite/magnetite

interface with ultrafine and large grain sizes. The BSE-SEM images (Fig. 7a, b) reveal a noticeable difference in the magnetite morphology that develops near the interface during the initial reduction of hematite to magnetite. In the large-grained sample the magnetite develops a distinctive "cell structure" with extremely fine nanopores surrounded by dense magnetite (near the magnetite grain boundaries), previously described in partially reduced single crystal hematite[26]. This morphology can be seen to persist in the wüstite as well (see Supplementary Fig. 12a, b). These "cells" are individual magnetite grains that grew within the hematite grains (see the EBSD phase and IPF maps in Supplementary Fig. 12c, e). The densely-packed nanopore channels do not develop at the "fresh" magnetite grain boundaries, creating this distinct morphology. Conversely, in the ultrafine-grained sample this morphology is not discernible, as the original hematite grains are too small for multiple magnetite cells (grains) to nucleate and grow within the same grain. This is determined by the minimum magnetite "cell" size observed in the large-grained sample (~1 μm), which is the same as the average hematite grain size in the ultrafine-grained sample. Nevertheless, there appear to be some small (~1 μm) porous pockets surrounded by denser regions in the ultrafine-grained samples (Supplementary Fig. 12c, d).

Further characterization data including higher/lower magnification SEM images and larger EBSD maps are provided in Supplementary Figs. 13, 14 for the ultrafine- and large-grained hematite samples, respectively. In the large-grained samples, the magnetite further away from the hematite retains notable localized texture reminiscent of the preceding large hematite grains (Supplementary Fig. 14c, d), mostly

comprising subgrains or submicron grains roughly the same size as those found in the ultrafine-grained sample magnetite. Overall, these differences in the magnetite morphology for different grain sizes further serve to explain the considerable differences in direct reduction behavior, derived from the different pore networks generated in large or small grains of polycrystalline iron oxide.

## Conclusions

This study revealed the influence of hematite grain size on the hydrogen-based direct reduction of iron oxide. Dense sintered pure hematite samples with large (~30 μm) and ultrafine (~1 μm) grain size were obtained by air sintering (AS) and spark plasma sintering (SPS), respectively. These polycrystalline samples were utilized to isolate and assess the various effects of initial hematite grain size, as pertaining to the kinetics and microstructure evolution during hydrogen-based direct reduction (at 700 °C conducted in a TGA system).

It was found that finer-grained hematite reduces more slowly in the earlier stages, up to about ~40% reduction degree, but more quickly during the later and especially final stages of reduction (10–20% reduction degree). Notably, at slow heating rates (2 °C/min) both types of samples were fully reduced (>99%), while at fast heating rate (20 °C/min) the large-grained hematite reduces less effectively (~86%) than the ultrafine-grained samples (~93%), retaining considerably more unreduced oxides.

The microstructure analysis of partially reduced samples revealed various noteworthy insights into microstructure evolution during direct reduction:

- Larger hematite grains enable well-connected, extremely fine directional pore channels to form in the magnetite near the hematite/magnetite interface, while small grains imply a multiplicity of orientations and thus the disruption of the fine pore network. As a result, small grains induce formation of a coarser but more homogeneous pore network, which is more effective in the later stages of reduction.
- The heterogeneity of the pore network in large-grained samples increases susceptibility to retaining unreduced oxides encased by dense metallic iron.
- Based on the observations made for annealed SPS samples with intermediate grain sizes, there is a clear trend that pore channels are finer with increasing grain size and the threshold grain size for the formation of straight directional pores is somewhere between 5 and 10 μm.
- Large grains produce an initial magnetite "cell" morphology, previously observed during reduction of single crystal hematite, due to the nucleation and growth of multiple magnetite grains in a single large hematite grain, which cannot develop in ultrafine grains.
- In the large-grained hematite, a sizeable fraction of the magnetite formed via S-N OR (~40% of the phase interface), retaining highly textured large magnetite grains containing {111} 60° FCC twins. In contrast, in regions without the S-N OR the initial magnetite was substantially refined (<1 μm grain size) and untextured.
- The iron formed by reduction of the large-grained hematite had a larger average grain size (~0.56 μm) and included relatively large irregular grains containing low-angle grain boundaries, in contrast to the finer (~0.34 μm) and more equiaxed grains in the iron formed from ultrafine hematite grains.

In conclusion, larger hematite grain size results in faster reduction kinetics during the early stages, while there is advancement of a pronounced reduction front facilitated by pore channel formation. However, at the late stages of reduction, the more homogenous and coarser pore network created in fine-grained samples enables faster and more effective reduction, making it easier to achieve a higher degree of metallization.

The findings of this study elucidate how the grain size of the precursor phase predominantly influences the solid-state direct reduction processes by dictating the pore network formation. Although discussed here principally in the context of reduction of iron oxide for metal production, the microstructure evolution insights can be useful for designing porous materials for various applications such as catalysts or electrodes, as well as shedding light on microstructural degradation mechanisms during cyclic redox processes.

## Methods

### Polycrystalline hematite sample preparation

Dense polycrystalline hematite samples were prepared by sintering submicron hematite powder (Sigma Aldrich; ≥96% pure, ≤5 μm particle size). SEM examination (Supplementary Fig. 1) revealed submicron powder with a particle size of ≤1 μm. The ceramic grain size was controlled by means of the sintering or annealing procedure. To produce samples with a large grain size (34.5 ± 14.7 μm), the hematite powder was compacted into green bodies and pressureless sintered in air inside a muffle furnace at 1200 °C for 4 h with a heating and cooling rate of 2 °C/min; these samples were termed air sintered (AS). Spark plasma sintering (SPS) was employed to minimize grain growth and retain an ultrafine grain size (1 ± 0.4 μm). The sintering parameters consisted of heating at 25 °C/min to 900 °C and holding for 30 min under an applied pressure of 63 MPa. Note that the SPS temperature was low enough to limit most of the reaction between iron oxide and the graphite foils and die. Nevertheless, a thin outer layer of magnetite (~200 μm) readily formed, also causing the sample to fracture into several pieces during the cooling period. All the magnetite at the surface was thoroughly removed by grinding prior to the reduction experiments. The SPS samples also contained a minor fraction of magnetite phase (<4%, see Supplementary Fig. 3) due to the starting powder containing some magnetite, which could not oxidize to hematite as in the AS sample. Furthermore, to obtain samples with intermediate grain sizes, some of the sintered samples were then heat treated at 1000 °C for 6 h or 1200 °C for 12 h yielding hematite with ~5 μm and ~10 μm grains, respectively. To negate any sample geometry or size effects on the direct reduction process, cylindrical samples with nearly identical dimensions were prepared. The samples were fashioned into cylindrical specimens by grinding to a height of ~3 mm and subsequent machining by a lathe to a diameter of ~5 mm. The weight of the dense iron oxide specimens prior to reduction was ~0.28–0.31 g.

### Hydrogen-based direct reduction experiments

Hydrogen-based direct reduction was carried out in two different kinds of furnaces. The experiments to study the reduction kinetics were carried out for two samples with the extreme (large ~30 μm and ultrafine ~1 μm) grain sizes using a commercial thermogravimetric analysis (TGA) furnace (TA instruments DynTHERM). For each run a sample was placed in an alumina crucible hung by a platinum wire and the weight changed was continuously monitored at ~2 s intervals. The experiments were performed at a temperature of 700 °C with a heating rate of 10 °C/min under a pure $H_2$ gas flow at a rate of 10 L/h followed by isothermal holding for 30 min at 700 °C. Note that the selection of a temperature of 700 °C in this study serves as a representative intermediate direct reduction temperature, typically used in microstructural studies, to slow down kinetics into an observable range.

Partial reduction for microstructural analysis of the samples with different grain sizes was carried out in a custom-made setup consisting of a vertical quartz tube infrared furnace[68], supporting an open quartz basket enabling placement of multiple samples together. The samples with four types of grain sizes (~1, ~5, ~10, and ~30 μm) were reduced simultaneously thereby having them subject to identical reducing conditions. The partial reduction was performed using the same parameters used for the kinetics measurements consisting of the same temperature (700 °C), heating rate (10 °C/min) and gas flow (pure $H_2$, 10 L/h) but held for only a short duration of 1 min at 700 °C.

### Microstructure characterization

The density and porosity of the different sintered samples was measured before and after reduction using the Archimedes principle. The partially reduced samples were cut along their round cross-section using a diamond wire saw. Pieces of the sample exposing *i)* reduced surface and *ii)* the cross-sections containing the reduced outer layer were embedded in conductive resin and metallographically prepared by grinding with SiC abrasive papers, polishing with 3 μm diamond suspension, and a final step of silica

nanoparticle suspension (OP-S). The samples exposing the reduced surface were ground until the hematite/magnetite interface was reached. The microstructures were examined using a Zeiss Sigma 500 high-resolution scanning electron microscope (SEM) operated at 15 kV and 9.5 nA. The analysis was carried out using backscattered electrons (BSE) signal and electron backscatter diffraction (EBSD) collected using an EDAX Hikari Super camera. The obtained Kikuchi patterns were reindexed by spherical indexing[69] using OIM Matrix v.9 software. A series of sequential BSE-SEM images of the reduced layer in partially reduced samples were used to assess the oxide/metal phase and porosity area fractions as a function of the distance from the hematite/magnetite interface in the ultrafine and large grain samples by image analysis using Fiji ImageJ software v1.54p.

## Data availability

The datasets generated during and/or analyzed during the current study are available from the corresponding author on reasonable request.

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

## Acknowledgements

B.R. acknowledges the support of an Alexander von Humboldt Foundation Fellowship (hosted by D.R.). M.R. acknowledges the financial support of a Max-Planck-Gesellschaft Scholarship. Y.M. acknowledges financial support offered through the Walter Benjamin Programme of the Deutsche Forschungsgemeinschaft (Project No. 468209039). Y.M. discloses support for the research of this work from Horizon Europe project HAIMan co-funded by the European Union grant agreement (ID 101091936). D.R. discloses financial support from the European Union through the ERC Advanced grant ROC (Grant Agreement No. 101054368).

## Author contributions

B.R. conceptualized the study, fabricated the samples, performed materials characterization and analysis, and prepared the initial draft. M.R. performed microstructure characterization and crystallographic analysis. S.S. performed the TGA experiments and data analysis. Y.M. and D.R. provided resources. All co-authors contributed to the discussion and manuscript review and editing.

## Funding

## Competing interests

The authors declare no competing interests.
