## [Transparent Peer Review file · Communications Materials]

Influence of grain size on the solid-state direct reduction of polycrystalline iron oxide

Corresponding Author: Dr Barak Ratzker

Version 0:

Decision Letter:

**** Please ensure you delete the link to your author homepage in this email if you wish to forward it to your coauthors ****

Dear Dr Ratzker,

Thank you again for submitting your manuscript "Influence of grain size on the solid-state direct reduction of polycrystalline iron oxide" to Communications Materials. We have now received reports from 3 reviewers and, based on their comments, we have decided to invite a revision of your work. You will find the reviewers' reports below. While they find your work of interest, they have raised important points which must be addressed in a revised manuscript.

To allow us to move forward with your work, we also ask that you edit your manuscript according to the attached table.

Please read this document carefully as we will be unable to further assess your revised paper until these important points are addressed.

Please outline all revisions made in the right-hand column and return the completed table with your updated manuscript files as a Related Manuscript file.

When resubmitting, please also include:

- A point-by-point response to the reviewers' comments. If you are unable to address specific reviewer requests or find any points invalid, please explain
- A clean version of your revised manuscript with no mark-ups
- A marked-up version of your paper with all changes highlighted in a different colour

Please use the link below to submit your revised files:

Link Redacted

**** This url links to your confidential home page and associated information about manuscripts you may have submitted or that you are reviewing for us. If you wish to forward this email to co-authors, please delete the link to your homepage first ****

We hope to receive your revised paper within six weeks, but we understand that the revisions may take longer. Please let us know if you find that the revision process will take substantially more time.

We are committed to providing a fair and constructive peer-review process. Please do not hesitate to contact me if you have any questions or would like to discuss these revisions further. We look forward to seeing the revised manuscript and thank you for the opportunity to review your work.

Best regards,

John Plummer, PhD
Chief Editor
orcid.org/0000-0003-4824-8497
Communications Materials

Reviewers' comments:

Reviewer #1 (Remarks to the Author):

They investigate the influences of grain size on the solid-state direct reduction of polycrystalline iron oxide using TGA, microstructure, and EBSD analysis.

They claim that the change in reduction rates is related to pore networks. However, the most experimental results are obtained in 2D, and there is little information on the development of pore networks in 3D, which is essential to understand the phenomena. Moreover, quantitative modeling or analysis is not discussed in the paper.

They have been reported by previous studies such as:

T. Wolfinger et al. (2022), 10.1007/s11663-021-02378-1

T. Takayama et al. (2025), 10.1016/j.actamat.2025.121470

So I think that it is difficult to publish the manuscript in this journal in its present form.

Reviewer #2 (Remarks to the Author):

Article title: Influence of grain size on the solid-state direct reduction of polycrystalline iron oxide

Reviewer Recommendation Term: Major Revision

This thesis systematically investigated the influence of hematite grain size on its solid-state direct reduction behavior in hydrogen. The topic has clear academic value and industrial application background. Through the preparation of dense polycrystalline hematite samples with different grain sizes (approximately 1 μm and 30 μm), combined with thermogravimetric analysis and multi-scale microstructure characterization, the significant impact of grain size on reduction kinetics, pore evolution, and phase transformation behavior was revealed. After a comprehensive review, the thesis requires major revisions. The following are the revision suggestions:

1. Why did you choose this temperature of 700 °C? The selection of temperature is of vital importance for the evolution of grain size. It is recommended to provide the basis for this temperature selection.
2. Some supplementary information is frequently cited in the main text, indicating that they are crucial data. Therefore, they should be included in the main text and a concise summary of the key findings should be provided.
3. There are significant differences in the preparation methods of large-particle samples and ultrafine-particle samples. Can the difference in the final reduction degree of the two be simply attributed to whether the grain size is reasonable?
4. During the raw material preparation stage, the basic physical parameters of the initial iron ore powder should be provided, including specific surface area, particle size distribution and apparent morphology, etc.
5. The microscopic structure and phase analysis of the samples were conducted after partial reduction for 1 minute or complete reduction for 30 minutes. Could you add intermediate reduction data to determine the changes in pores and grain boundaries during the reduction process?
6. Some parts of the content are not written properly. For example, for SPS, only the full name and abbreviation need to be written the first time it is mentioned, and then the abbreviation can be used subsequently.
7. This manuscript states that large particle samples will form directional nanochannels, while ultrafine particle samples will form spherical coarse pores. This alone is not sufficient to explain the situation. Has there been any supporting evidence from existing literature or theoretical calculations?

Reviewer #3 (Remarks to the Author):

The investigation aims to uncover the effect of grain size on direct reduction behavior of hematite by focusing on a comparison between two sintered samples with considerably different grain sizes of ~30 μm and ~1 μm . The manuscript presents very thorough characterization work; additional details on the characterization methodology would further strengthen the paper. The paper is very well-written and requires no language modification.

The main points detracting from the strength of the paper are:

1. The authors stated that although some past studies have erroneously equated 'grain size' with pellet or powder particle size, the specific effect of grain size on the direct reduction of polycrystalline iron oxide has not yet been examined. Include the initial porosity value of the pellets? Was there a big variation in the initial porosity value prior to reduction tests? Distinguishing between the grain size and particle size is an important point. It would be helpful to cite studies in which 'grain size' was erroneously equated with pellet or powder particle size, so that this statement is clearly grounded in the existing literature.
2. Sintering time and temperature influence the pore size distribution. Has the pore size distribution of the two samples been characterized and compared? The potential difference in pore size distribution could have influenced the reduction behavior.
3. How consistent was the porosity of the samples sintered with the same method. Different initial porosities could affect the reduction behavior using various heating rates.

4. Although the isothermal heating time was constant at different heating rates, high heating rate implies a shorter cycle in total. Could this contribute to higher difference in reduction degree between the two samples at higher heating rates (fig. 1.e)?
5. Beyond ~60 μm from the hematite/magnetite interface, the porosity fraction in the ultrafine-grained sample is higher than large-grained samples. Please provide potential reasons for this difference. Since porosity formation (regardless of its directionality) is a result of reduction, one would expect to see higher iron in fine-grained sample accompanied by higher porosity. However, observations show lower Fe accompanied by higher porosity fraction.
6. Please describe how the porosity fraction is quantified and what is the minimum detectable porosity size in partially reduced samples?
7. What is the selection
8. It is noted that given the microstructural similarities between H₂-DRI and conventional DRI, the influence of grain size will be comparable for other gaseous reductants like CO or CH₄. This statement is not accurate, specially because pore coarsening is possible in CO reduction because of the slower reduction. In addition, carbon deposition could change pore accessibility.
9. Why doesn't nucleation of multiple magnetite grains in single hematite grain occur in fine-grained samples? It is hypothesized that it is due to the small grain size in these samples. Is there a theoretical justification supporting this hypothesis? Alternatively, could this observation simply result from insufficient resolution to distinguish multiple ultra-small magnetite grains forming in the fine-grained sample?

** See Nature Research's author and referees' website at www.nature.com/authors for information about policies, services and author benefits

Version 1:

Decision Letter:

** Please ensure you delete the link to your author homepage in this email if you wish to forward it to your coauthors **

Dear Dr Ratzker,

Thank you once again for submitting your manuscript, "Influence of grain size on the solid-state direct reduction of polycrystalline iron oxide," to Communications Materials. Your manuscript has been seen again by the referees, whose comments are appended below. I am happy to say that the concerns of our reviewers have now been addressed, and that we only require some minor amendments before we can accept your paper.

Our remaining requests are:

-If data are reported in the study, the statement should specify, at a minimum, that all relevant data are available from the authors upon request. This statement should include details on who will be responsible for replying to this request along with the email address.

This will be the final revision of your manuscript. We ask that you carefully review all files associated with your paper and follow the link below to upload the final version of all files, including display items and supplementary material. Please ensure that these files are clean, without any markups or comments, as they will be sent for publication.

Link Redacted

We hope to receive this updated version of your paper within 1 week, but please let us know if you find that you need more time.

We hope to receive this updated version of your paper **within 1-week**, but please let us know if you find that you need more time.

Best regards,
John Plummer, PhD
Chief Editor
orcid.org/0000-0003-4824-8497
Communications Materials

Reviewers' comments:

Reviewer #1 (Remarks to the Author):

Thank you for your reply to my comments. I understand your point that a discussion based on 2D observations is enough to understand the reduction mechanism in 3D. However, the limitation of 2D observations remains important for understanding the reduction mechanism, which proceeds differently along 3D or crystallographic directions, as discussed in the references cited in this paper. And this point should be commented on in the manuscript. The final decision on publication should be considered based on this point and the reply to comments by the other referees.

Reviewer #3 (Remarks to the Author):

The authors have addressed the comments in the revised version.

Version 2:

Decision Letter:

Dear Dr Ratzker,

We are delighted to accept your manuscript titled "Influence of grain size on the solid-state direct reduction of polycrystalline iron oxide" for publication in Communications Materials. Thank you for choosing to publish your interesting work with us.

Acceptance of your manuscript is conditional on all authors' agreement with [our publication policies](https://www.nature.com/commsenv/editorial-policies). In particular, your manuscript must not be published elsewhere and there must be no announcement of the work in the media until the publication date.

Please carefully read the information below for information on what to expect from the next steps of the publishing process:

Article in Press:

Please note that in advance of your paper being published we will host an early access version, known as an 'Article in Press,' on our journal website. For more information on this initiative please see our [author guidelines](https://support.springernature.com/en/support/solutions/articles/6000281821-what-is-an-article-in-press-).

Publication as an [Article in Press](https://support.springernature.com/en/support/solutions/articles/6000281821-what-is-an-article-in-press-) is typically within 1-2 weeks after we have received your corrected proofs and publishing agreement. Subsequently, we will aim to publish the Version of Record in a timely manner. Please note there will be no further correspondence about your publication date.

When your article is published as the Version of Record, you will receive a notification email. **If you are planning an embargoed press release or require a specific publication date, please complete our [scheduling requests form](https://forms.office.com/e/ed7NBDd08u), or contact commsproduction@springernature.com, as soon as possible after acceptance and we will endeavour to accommodate your request.**

For further information on the journey of your article from acceptance to publication, please see our [Author FAQs](https://www.nature.com/documents/Author_FAQs.pdf).

Publishing Agreements and Fees:

In about one week, you will receive an email with a link to complete the appropriate grant of rights necessary for publishing your paper and – if applicable – to provide payment information for your article-processing charge (APC), either via credit card or by requesting an invoice.

If needed, our Author Services team will be in touch regarding any additional information that may be required.

In order to avoid any delays, please ensure that you have emails from Springer Nature whitelisted in your mail system.

Proofs:

At this point we will also edit your manuscript to ensure that it conforms with our house style. Once you have completed the publication agreement and arranged payment, you will receive a separate email with a link to an online eProof for you to review. Please read your proof with great care to ensure that no changes have been introduced which have inadvertently altered the meaning of your paper. We suggest that you discuss the proof with your co-authors, but please ensure that only one author communicates with us and that only one set of corrections is returned via the online correction in the eProof.

The corresponding (or nominated) author is responsible on behalf of all co-authors for the accuracy of all content, including spelling of names and current affiliations.

To ensure prompt publication, your proofs should be returned within two working days. If there is any period within the next four weeks in which you won't be available, please nominate a co-author with whom we can correspond and send their contact information to us via email at commsproduction@springernature.com as soon as possible.

Please note that your Supplementary Information files are now finalised, and they will be submitted as provided for preparation for publication of the Article. Any requests to make changes will only be considered in exceptional circumstances and will result in a delay to publication.

You will not receive access to your eProof until the Licence to Publish and Article-Processing Charge steps are completed.

We welcome the submission of material for the 'Featured Image' section of the Communications Materials home page. Images should relate to the content of your manuscript, but do not need to be taken directly from the accepted work. Suggestions should be sent by email to commsmat@nature.com, along with the completed [Licence to Publish](https://resource-cms.springernature.com/springer-cms/rest/v1/content/18943626/data/Research-Permission-Template-EN) form. Please note that images should be supplied as 1400x400-pixel, in RGB. Unfortunately, we cannot promise that your suggestions will be used.

Providing great service is very important to us. We would greatly appreciate any comments you have about your experience at Communications Materials. We look forward to publishing your paper, and we hope to work with you again in the future.

Best regards,
John Plummer, PhD
Chief Editor
orcid.org/0000-0003-4824-8497
Communications Materials

*You can now use a single sign-on for all your accounts, view the status of all your manuscript submissions and reviews, access usage statistics for your published articles and download a record of your reviewing activity for the Nature Portfolio journals. Please check your account regularly and ensure that we have your current contact information.

We may promote your article on social media once it is published, so please feel free to send me the twitter handles of any authors or departments and we will be sure to tag them accordingly.

Response to reviewer comments

We would like to express our sincere gratitude to the editor for handling this manuscript and to all the reviewers for taking their time to thoroughly review our manuscript and for the constructive feedback and suggestions. In the following, we provide a point-by-point response report containing and explaining in detail all revision items. Find our responses in **blue** and relevant modifications in the manuscript or supplementary information highlighted in **yellow**.

Reviewer #1

Overall comment

They investigate the influences of grain size on the solid-state direct reduction of polycrystalline iron oxide using TGA, microstructure, and EBSD analysis.

They claim that the change in reduction rates is related to pore networks. However, the most experimental results are obtained in 2D, and there is little information on the development of pore networks in 3D, which is essential to understand the phenomena. Moreover, quantitative modeling or analysis is not discussed in the paper.

They have been reported by previous studies such as:

T. Wolfinger et al. (2022), 10.1007/s11663-021-02378-1

T. Takayama et al. (2025), 10.1016/j.actamat.2025.121470

So I think that it is difficult to publish the manuscript in this journal in its present form.

Overall response

We highly appreciate the reviewer for kindly taking the time to review our manuscript, and we hope that the following explanations are sufficiently convincing as to why these are not issues hindering publication of our study.

The claims regarding different reduction rate and behavior depending on grain size are supported by the empirical evidence. The vast difference in behavior between the samples with different grain sizes is the development of pore networks being influence by the presence of boundaries and texture changes which would agree with fundamental metallurgy concepts related to diffusion void formation. We believe that claiming that the experimental results are insufficient because the microstructure was analyzed only in 2D is incorrect for the following reasons: (1) Our analysis of pore morphology combines images from two different sample directions, allowing to assess porosity developed both in the round cross section (perpendicular to the advancing front) and in parallel to the reduction direction, which is an approach that enables us to infer the 3D structure (similarly to how we conducted microstructural examination in our previous work on single crystal hematite <https://doi.org/10.1016/j.actamat.2025.121174>). (2) Furthermore, in any given 2D image we have a multiplicity of orientations (note that the hematite polycrystalline samples are isotropic without any initial discernible texture) giving statistical validity to the observations related to pore morphology. (3) While true 3D analysis would be desirable, methods like micro-CT are not sensitive enough to detect nanoscale pores (typical resolution in the micron to submicron range); other methods such as FIB tomography have been attempted, but the combination of relatively high porosity fractions with such

fine submicron and nanoscale channel sizes causes the pores to erode during milling, such that they coarsen and lose their original shape, leading to artifacts in the imaging data and making it an unviable analysis method. (4) It is common practice to assess pore morphology from 2D images, as pore and pore channels generally have a spherical (or somewhat oval) cross-section; analyzing 2D images from multiple sample directions is thus meaningful, and averaging the analysis over a large region allows us to represent the overall porosity morphology and fraction accurately.

Regarding the lack of quantitative modelling, there are no fully quantitative models that are particularly relevant to this work. If the reviewer is referring to the common models used to assess reduction kinetics of DRI pellets like the shrinking core and pore/grain size models, these do not account for initially dense samples and are not practical for dense model systems as we use here to isolate the effect of grain size on direct reduction. To the best of our knowledge there are no quantitative models for direct reduction that have a component or variable related to grain size. To develop a new model that can fully capture this grain size effect with full consideration of all redox processes, phase transformations and the associated elastic – plastic response under full consideration of the correct thermodynamic potentials is unfortunately out of the scope for this work. Finally, we do not see how the suggested papers undermine the novelty or significance of our study. There are numerous works on iron oxide reduction (as cited in our literature review), but none so far (including these two papers) uncover the effect of grain size on the direct reduction process.

-End of the response to Reviewer #1-

Reviewer #2

Overall comment

Article title: Influence of grain size on the solid-state direct reduction of polycrystalline iron oxide
Reviewer Recommendation Term: Major Revision

This thesis systematically investigated the influence of hematite grain size on its solid-state direct reduction behavior in hydrogen. The topic has clear academic value and industrial application background. Through the preparation of dense polycrystalline hematite samples with different grain sizes (approximately 1 μm and 30 μm), combined with thermogravimetric analysis and multi-scale microstructure characterization, the significant impact of grain size on reduction kinetics, pore evolution, and phase transformation behavior was revealed. After a comprehensive review, the thesis requires major revisions. The following are the revision suggestions:

Overall response

We appreciate the reviewer for acknowledging the significance of our work and for the insightful questions and suggestions.

Comment 1

Why did you choose this temperature of 700 °C? The selection of temperature is of vital importance for the evolution of grain size. It is recommended to provide the basis for this temperature selection.

Response 1

The reviewer rightfully states that the choice of reduction temperature is crucial as it significantly influences reduction behavior. We had chosen here to conduct a comprehensive analysis of the influence

of grain size at 700 °C for two main reasons: (1) It serves as a representative intermediate reduction temperature; low enough so that differences in reduction kinetics become noticeable but high enough that reduction mechanisms and microstructure features are similar to technologically relevant temperatures. Furthermore, (2) it enables direct comparison to our previous studies on iron oxide performed at 700 °C using the same experimental systems, most notably our recent study on hematite single crystal (<https://doi.org/10.1016/j.actamat.2025.121174>). We wanted to execute a targeted and focused study where all conditions but the grain size of iron oxide polycrystalline samples are kept the same.

As suggested by the reviewer, we fully comply and have added a brief explanation for the rationale of performing the reduction experiments at 700 °C in the Methods section.

On a final note, indeed, the grain-size dependence of the reduction behavior could be influenced by the reduction temperature, but it is out of scope for this study and would be of interest to look into in future studies.

Additions to the main text:

Note that the selection of a temperature of 700 °C in this study serves as a representative intermediate direct reduction temperature, typically used in microstructural studies, to slow down kinetics into an observable range.

Comment 2

Some supplementary information is frequently cited in the main text, indicating that they are crucial data. Therefore, they should be included in the main text and a concise summary of the key findings should be provided.

Response 2

Thank you for this comment. We have considered this suggestion, but upon review of the manuscript, we did not find instances where it would be necessary to incorporate the supplementary figures in the main text, as this can lead to a detraction from the flow presenting the results and to overloading the main text. The supplementary data is complementary and mentioned in the appropriate places where they pertain to the discussion in the manuscript, to encourage the readers to find this additional information, in case they wish to delve deeper into methods and collected data. We believe this is the best practice to improve the focus and readability of the paper by relegating non-essential but useful data to the supplementary information.

Comment 3

There are significant differences in the preparation methods of large-particle samples and ultrafine-particle samples. Can the difference in the final reduction degree of the two be simply attributed to whether the grain size is reasonable?

Response 3

Indeed, there are significant differences in the preparation methods, which is what makes it possible to investigate such model systems with significantly different grain sizes. The main point of the paper is that we tried (as much as possible) to use samples that are identical in everything but the grain size (e.g., starting from the same powder, minimal porosity, same sample geometry, etc.), so that the only significant factor that can generate differences in the iron oxide reduction behavior (i.e., reduction kinetics and microstructure evolution) originates from the differences in grain size. The initial conditions of the samples with different grain sizes were compared (e.g., Table 1 and Supplementary

Figures 2 and 3) to emphasize this and rule out other possible factors. Therefore, the notable differences in final reduction degrees with heating rates provides a clear indicator that there are pronounced differences in reduction kinetics depending on initial hematite grain size.

Comment 4

During the raw material preparation stage, the basic physical parameters of the initial iron ore powder should be provided, including specific surface area, particle size distribution and apparent morphology, etc.

Response 4

We did not conduct a thorough characterization of the iron oxide powder since this was not a study about sintering, and only the resulting iron oxide polycrystalline ceramics final state (i.e., grain size, density) truly matter to answer the research question. While these technical details about the initial powder would be important for a study revolving around the sintering process or comparing the different behavior for different sintering methods, here we only use two sintering methods as means to produce pure polycrystalline iron oxide samples with distinctly different grain sizes. The powder is commercial iron oxide from a known manufacturer (already detailed in the manuscript methods section); well-suited for research with fine particle size and spherical morphology making them ideal for sintering. Since the specific surface area and particle distribution are not declared by the manufacturer we alternatively opted to examine them by SEM and add images showcasing the submicron powder morphology to the supplementary information (Supplementary Figure 1).

Addition to the main text:

Sintered hematite samples were produced using submicron hematite powder (Supplementary Figure 1).

Addition to the methods text:

SEM examination (Supplementary Figure 1) revealed the powder is submicron with a particle size of $\leq 1 \mu\text{m}$.

Additions to the supplementary information:

Supplementary Figure 1 | Morphology of hematite powder. BSE-SEM images showing the morphology of the hematite powder used for produced the sintered samples; (a) low and (b) high magnification. The particles are mostly spherical with a particle size $\leq 1 \mu\text{m}$.

Comment 5

The microscopic structure and phase analysis of the samples were conducted after partial reduction for 1 minute or complete reduction for 30 minutes. Could you add intermediate reduction data to determine the changes in pores and grain boundaries during the reduction process?

Response 5

Thank you for this comment. First, we should clarify that microstructural examination was conducted only on partially reduced samples, and the complete reduction experiments were only used to assess the reduction kinetics and total conversion rates from the weight change. The rationale behind this methodology is that for dense samples, partially reduced samples provide a meaningful viewpoint encompassing various microstructures representative of practically all stages of reduction. The microstructural progression of the reduction process can be inferred from examination of the reduction front leading out to the surface of the sample: the region near the hematite core, where magnetite is being created, shows the first reduction step and the related creation of porosity, while the reduced layers closer to the surface show the later, iron-rich stages. If intermediate or more advanced stage samples are examined, these show the same microstructure, only with different spatial distribution, as the majority of the sample resembles the reduced outer iron-rich regions. We have shown this very clearly in our previous study on iron oxide single crystal samples (<https://doi.org/10.1016/j.actamat.2025.121174>). Therefore, we can be certain that the microstructures observed gradually along the reduction front in partially reduced samples are sufficient to identify and elucidate the microstructure-dependent direct reduction mechanism, and specifically the effect of grain size herein.

Comment 6

Some parts of the content are not written properly. For example, for SPS, only the full name and abbreviation need to be written the first time it is mentioned, and then the abbreviation can be used subsequently.

Response 6

Thank you for this suggestion. The AS and SPS abbreviations were purposefully mentioned in full in three different places. Specifically, first in the main text as usual, second in the conclusions to provide a short summary of the work that might stand alone, and third in the methods section since it is somewhat separate and secondary to the main text (in journals like *Communications Materials* that follow this format). In all other instances the abbreviations were used. We think this helps improve readability of the paper, especially since they are not commonly used in the context of this field. If instructed to change this during the editorial process we will be happy to comply.

Comment 7

This manuscript states that large particle samples will form directional nanochannels, while ultrafine particle samples will form spherical coarse pores. This alone is not sufficient to explain the situation. Has there been any supporting evidence from existing literature or theoretical calculations?

Response 7

We are not sure what “situation” is being referred to, whether it is the general difference in reduction behavior depending on grain size or specifically the pore formation in large or ultrafine grains. In any case, we provide strong evidence that this is indeed the case. We ensured that samples were similar in

all but grain size and performed careful experiments with the microstructures that were analyzed produced under identical reduction conditions. Our conclusions follow strict logic, where frequent texture changes prevent the formation of longer and more directional channels that can be created by vacancy accumulation and void coalescence, depending on preferred crystal planes and/or directions. Given that our previous work on single crystal hematite without any grain boundaries presents similar pore formation mechanisms to those in the larger grains (which locally mimic smaller single crystals), our findings are further supported.

Previous studies provide sufficient microstructural evidence showing that our general pore formation observations in dense iron oxide are characteristic of the process. This is something we have now further emphasized in the beginning of the discussion on the pore networks to establish a stronger basis for the ensuing comparison based on grain size (see main text addition). The influence of grain boundaries and texture changes on different scales has not been examined or even explicitly considered so far, hence the novelty of our work.

We are not aware of theoretical calculations pertaining to this problem that we could conduct, as current and accepted models are used to describe the reduction behavior on porous pellets at a macroscopic scale. If the reviewer is referring to simulations, such as phase field modelling, these are extremely complicated to execute and beyond the scope of this paper, while current porosity modelling works are limited in predicting pore development and especially inadequate for the nuances of nanoscale or submicron channel formation and architectures. Although it would be ideal to have theoretical work or simulations to back up our findings, the substantial and comprehensive experimental evidence is sufficient to establish clear conclusions.

Addition to the main text:

Note that the formation of fine interconnected pore networks in magnetite is a hallmark of direct reduction^{9,22-24}, facilitating the process in dense material.

-End of the response to Reviewer #2-

Reviewer #3

Overall comment

The investigation aims to uncover the effect of grain size on direct reduction behavior of hematite by focusing on a comparison between two sintered samples with considerably different grain sizes of ~30 μm and ~1 μm . The manuscript presents very thorough characterization work; additional details on the characterization methodology would further strengthen the paper. The paper is very well-written and requires no language modification.

Overall response

We thank the reviewer for acknowledging the quality of our manuscript and the thoroughness of the characterization we performed as well as the insightful comments and useful suggestions that helped improved the manuscript.

Comment 1

The authors stated that although some past studies have erroneously equated ‘grain size’ with pellet or powder particle size, the specific effect of grain size on the direct reduction of polycrystalline iron

oxide has not yet been examined. Include the initial porosity value of the pellets? Was there a big variation in the initial porosity value prior to reduction tests?

Distinguishing between the grain size and particle size is an important point. It would be helpful to cite studies in which ‘grain size’ was erroneously equated with pellet or powder particle size, so that this statement is clearly grounded in the existing literature.

Response 1

To indicate the initial porosity values of our samples the relative density values (about 98% for AS and >99% for SPS) are given in Table 1 and their morphology in Supplementary Figure 2. Hence, both samples have very low initial porosity encompassing only isolated pores that do not affect the reduction process.

Regarding studies that mistakenly refer to particles size as grain size, we fully agree with the reviewer that it is important and therefore we have now added two relevant citations in association with this statement. Furthermore, this suggestion inspired us to clarify another critical point so that grains in our work are not mistaken for “grains” that describe isolated core-shell units during direct reduction as they are defined according to the ‘grain models’ used to assess reduction kinetics.

Addition to the main text:

It must also be clarified that herein the term ‘grains’ only refers to the crystallites comprising the polycrystalline oxide outlined by grain boundaries – and not what is sometimes in the literature referred to as “grains” in direct reduction kinetics grain models^{20,21}, which are the localized core-shell units that develop during reduction consisting of an oxide core enveloped by lower oxide states or metal and surrounded by large pores.

Comment 2

Sintering time and temperature influence the pore size distribution. Has the pore size distribution of the two samples been characterized and compared? The potential difference in pore size distribution could have influenced the reduction behavior.

Response 2

We appreciate the reviewer’s concern regarding initial sinter porosity, and we fully agree that it is true that initial porosity can indeed influence the reduction behavior, especially the overall kinetics. However, this is not a concern in our case as we readily obtained dense ceramics with very low porosities (~0.5–2 %) otherwise the sintering process would have been optimized further. This limited porosity consists of isolated pores situated at grain boundaries or within grains (for the air-sintered samples). Therefore, it is safe to assume that these isolated pores do very little to improve the pore network percolation and have a negligible effect on the reduction behavior.

To clarify this in the manuscript, we mention that Supplementary Figure 2 also shows the initial porosity and added a statement regarding the negligible effect that the isolated sinter pores would have on the reduction behavior.

Addition to the main text:

The initial porosity in all sintered samples was very low ($\leq \sim 2\%$), so that the isolated sinter pores have a negligible effect on the reduction behavior.

Comment 3

How consistent was the porosity of the samples sintered with the same method. Different initial porosities could affect the reduction behavior using various heating rates.

Response 3

Thank you for this comment. It is important that the porosity of samples was consistent, those with similar grain size (i.e. same sintering method) had similar porosity – validated by density measurements and microstructural inspection of multiple samples. We methodically prepared samples that were as identical as possible, both in their compaction and sintering process to produce consistent microstructures, and with regard to sample geometry when machining them into cylinders.

Comment 4

Although the isothermal heating time was constant at different heating rates, high heating rate implies a shorter cycle in total. Could this contribute to higher difference in reduction degree between the two samples at higher heating rates (fig. 1.e)?

Response 4

Thank you for this insightful question. The shorter reduction time for faster heating rates indeed contributes to the larger difference in reduction degree. The trend showing that the higher the heating rate the larger the difference in total reduction degree between the samples with markedly different grain sizes serves to emphasize the impact grain size has on the reduction. In industrially relevant scenarios heating rates may be ever faster, so this result critically points out that even in porous industrial pellets grain size could potentially have a noticeable effect on reduction kinetics.

Comment 5

Beyond ~60 μm from the hematite/magnetite interface, the porosity fraction in the ultrafine-grained sample is higher than large-grained samples. Please provide potential reasons for this difference. Since porosity formation (regardless of its directionality) is a result of reduction, one would expect to see higher iron in fine-grained sample accompanied by higher porosity. However, observations show lower Fe accompanied by higher porosity fraction.

Response 5

Thank you for another insightful question. At first thought it would seem logical that since porosity is created by oxygen removal as reduction progresses then also more iron should be present; however, that is only if one disregards the effects of sintering. The more iron is created, the more it sinters, compounded by the fact that since the channels are narrower in the large-grained sample, rapid sintering is promoted by stronger capillary forces and a higher proportion of surface to volume ratio (the driving force for sintering). Sintering is evident both in the microscale upon inspecting the microstructures in iron-rich regions and in the shrinkage of the sample post reduction (as shown in the before and after photographs in Figure 1d). To make this readily visible for the reader, we have prepared compound images that present nearly the whole reduced layer of both samples in sufficiently high magnification to clearly show the entire sequence of events from initial porosity in the magnetite and formation of the pore channels to the intermediate thin Fe layers and the denser iron-rich nearly fully reduced regions closer to the surface. These images can be found in the new Supplementary Figure 5. We have also added a sentence that explains this notion regarding sintering changing the porosity fraction in iron-rich regions.

Additions to the main text:

Supplementary Figure 5 presents compound SEM-BSE images of a slice encompassing nearly the entire reduced layer in both samples, showcasing the microstructure evolution sequence as reduction progresses.

Furthermore, the fact that the fraction of porosity is higher than iron in the large-grained sample and vice versa for the ultrafine-grained sample (Figure 3) can be explained by their different pore network morphology. The more metallic iron forming during reduction, the more it sinters, compounded by the fact that since the channels are narrower in the large-grained sample, rapid sintering is promoted by stronger capillary forces and a higher proportion of surface to volume ratio (the driving force for sintering).

Addition to the supplementary information:

Supplementary Figure 5 | Microstructure of the reduced layers. BSE-SEM collage images showcasing most of the reduced layer in the (a) large- and (b) ultrafine-grained samples. The orange arrows depict the continuation of the compound images.

Comment 6

Please describe how the porosity fraction is quantified and what is the minimum detectable porosity size in partially reduced samples?

Response 6

As mentioned in the manuscript, the porosity fractions were calculated by image analysis (using the ImageJ software threshold function). We used high-resolution SEM-BSE images with sufficiently high magnification (x5000) making it possible to detect pores down to roughly ≥ 20 nm in size. Examples of such the series of microstructure images are now given in the new Supplementary Figure 5 and examples of the porosity analyzed using the threshold function are shown in Supplementary Figure 7.

Comment 7

What is the selection

Response 7

This comment is unfortunately not complete so we cannot address it.

Comment 8

It is noted that given the microstructural similarities between H₂-DRI and conventional DRI, the influence of grain size will be comparable for other gaseous reductants like CO or CH₄. This statement is not accurate, specially because pore coarsening is possible in CO reduction because of the slower reduction. In addition, carbon deposition could change pore accessibility.

Response 8

We thank the reviewer for the critical thinking and for pointing out this issue. We agree that this statement was inadequate and relied on general observations in some studies without considering all relevant factors. Reduction by CO can induce different porosity fractions and pore morphologies, as well as cause carburization. Therefore, we chose to adjust the former statement, as it is unclear how similar grain size effects would be when using CO-based reductants.

Amendments to the main text:

Given the inconclusive similarities/differences in microstructure evolution observed for H₂ and CO reductants^{9,23,58,59}, it is unclear to what extent similar trends would occur when using other reducing gases like CO or CH₄.

Comment 9

Why doesn't nucleation of multiple magnetite grains in single hematite grain occur in fine-grained samples? It is hypothesized that it is due to the small grain size in these samples. Is there a theoretical justification supporting this hypothesis? Alternatively, could this observation simply result from insufficient resolution to distinguish multiple ultra-small magnetite grains forming in the fine-grained sample?

Response 9

We thank the reviewer for the careful reading and pointing out this detail. This made us realize that our description and wording were not precise enough. We emphasized nucleation while disregarding the

growth aspect. There can be multiple magnetite nucleation events within the same hematite grain, followed by their growth and eventual impingement on other grains. This may happen before magnetite grain expansion reaches the initial hematite grain boundaries, forming the magnetite “cell” structures mentioned in the manuscript. According to our findings the average minimum size of a magnetite “cell” is $\sim 1 \mu\text{m}$. As the average grain size in the ultrafine-grained sample is the same ($\sim 1 \mu\text{m}$), it then follows that such cells are not able to develop, as the nominal hematite grains are smaller than the critical size the magnetite grains reach before meeting other nucleated grains. The observation of some grain refinement in the ultrafine-grained samples (upon transformation to magnetite) indicates that multiple grains can nucleate even in small single hematite grains, but the “cell” morphology with nanoporous centers surrounded by dense walls is suppressed. Of course, this may depend on reduction temperature and other extrinsic factors, but for our case the reduction conditions were the same for both samples, so this empirical conclusion is well-founded. We have made some minor adjustments to the text to clarify this, specifically mentioning the growth factor which was not clearly mentioned before.

We don't think there is a significant characterization resolution issue. The step size of the EBSD analysis was 50 nm, and the spatial resolution allows to accurately distinguish submicron grains but limited for nanograins ($< \sim 100 \text{ nm}$).

Amendments to the main text:

These “cells” are individual magnetite grains that grew within the hematite grains (see the EBSD phase and IPF maps in Figure 12c,e). The densely-packed nanopore channels do not develop at the “fresh” magnetite grain boundaries, creating this distinct morphology.

Conversely, in the ultrafine-grained sample this morphology is not discernible, as the original hematite grains are too small for multiple magnetite cells (grains) to nucleate and grow within the same grain. This is determined by the minimum magnetite “cell” size observed in the large-grained sample ($\sim 1 \mu\text{m}$), which is the same as the average hematite grain size in the ultrafine-grained sample.

-End of the response to Reviewer #3-

Response to reviewer comments

We would like to once again express our sincere gratitude to the editor for handling this manuscript and to all the reviewers for taking their time to thoroughly review our manuscript and for the constructive feedback and suggestions. Find our response below (in blue) to the final reviewer comment and the additions to the revised text highlighted in yellow.

Reviewer #1

Overall comment

Thank you for your reply to my comments. I understand your point that a discussion based on 2D observations is enough to understand the reduction mechanism in 3D. However, the limitation of 2D observations remains important for understanding the reduction mechanism, which proceeds differently along 3D or crystallographic directions, as discussed in the references cited in this paper. And this point should be commented on in the manuscript. The final decision on publication should be considered based on this point and the reply to comments by the other referees.

Overall response

We highly appreciate the reviewer for the supportive comment. We agree that it is important to mention this point of using a 2D analysis to represent a 3D structure, in particular regarding the pore networks. Therefore, we have added a sentence including a relevant citation (Beckingham et al., 2D and 3D imaging resolution trade-offs in quantifying pore throats for prediction of permeability, *Advances in Water Resources* 62 (2013) 1–12) to emphasize that we are aware of the limitations regarding 2D microstructural analysis but that the observations should nevertheless be adequately representative. Furthermore, we also added another sentence in the discussion regarding the ground-in sample microstructures to clarify that it enables us to get a more accurate depiction of the 3D structure using this approach.

Additions to the main text:

Despite the inherent limitations of using 2D analysis to describe a 3D pore structure, the porosity fractions are assumed to be adequately representative⁶⁰.

This analysis is complementary to that conducted in cross-sectional view above, allowing for 3D understanding of the microstructure.

-End of the response to Reviewer #1-